# SeaDiff: Delve into Underwater Image Generation with Symmetrical Parameter Control

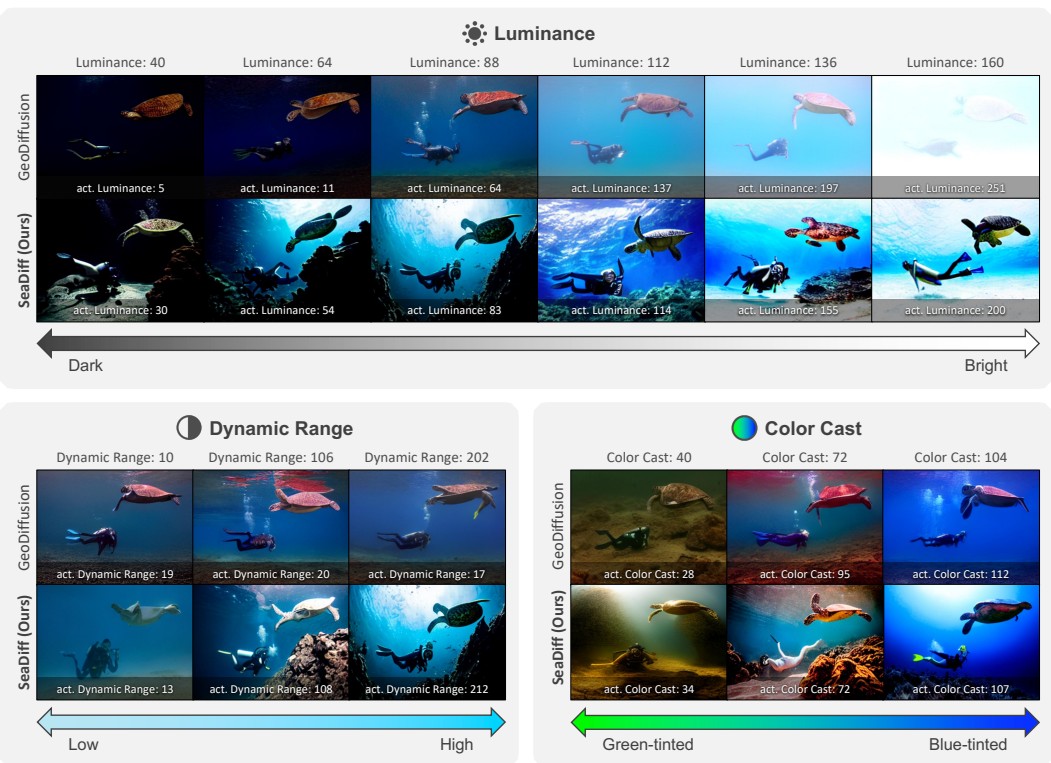

Figure 1: **Underwater image generation with precise appearance control.** Compared with existing methods, the proposed method, named SEADIFF, can precisely control the appearance of underwater images, such as luminance (top), dynamic range (lower left), and color cast (lower right). The text above each subfigure represents the input attribute to the model, while the label "act." on the images indicates the actual appearance attribute of the generated images. It can be observed that the underwater images, generated from left to right, show an increasing trend in appearance. Notably, these images were generated multiple times from scratch rather than being edited from a same image.

## Abstract

REBUTTAL20241120 With the advancement of diffusion models, the controllability of image generation has significantly improved. However, due to the refraction and absorption of light in water, underwater images often exhibit notable variations in luminance and color cast. This leads to challenges for generative models pre-trained on terrestrial images, as they struggle to produce underwater images with a diverse range of these variations, severely limiting the appearance diversity of generated underwater images. To address this issue, we focus on the precise control of appearance in underwater images. We model the appearance of underwater images using three attributes: luminance, dynamic range, and color cast. We propose a new method, SEADIFF, which introduces a Symmetrical Parameter Control structure to achieve precise control over the appearance of underwater images. The proposed method comprises two modules: Appearance Writer,

---

†Corresponding author.
Project Page:

which encodes and injects appearance attributes into the U-Net encoder, and Appearance Reader, which ensures that the generated images align with the desired appearance by analyzing the feature maps. Experimental results demonstrate that the proposed SEADIFF method significantly improves control over underwater image appearance while maintaining image quality, validating its effectiveness in underwater image generation.

# 1 INTRODUCTION

In the field of deep learning, data collection and annotation have always been challenging. This is especially true for underwater images. Due to the difficulties in collecting and annotating underwater data, existing underwater datasets (Fu et al., 2023; Jiang et al., 2021; Liu et al., 2021; Pedersen et al., 2019; Song et al., 2023a; Lian et al., 2023) often lack diversity. This is primarily reflected in the fact that many datasets are composed of similar frames extracted from the same video. As a result, these datasets exhibit low diversity in appearance such as luminance, dynamic range, and color cast, and typically only reflect the characteristics of specific water regions at similar depths. This lack of variability significantly limits the performance of underwater perception models (Xu et al., 2023; Yeh et al., 2021; Jia et al., 2022; Wang & Xiao, 2023). Meanwhile, the development of generative models has provided an effective means to expand existing datasets (Fang et al., 2024; Hao et al., 2024). Generative models have been widely used as a form of data augmentation (Zhang et al., 2024). A natural approach is to leverage these models to augment underwater datasets and mitigate the limitations imposed by the low diversity of existing data.

However, this approach does not yield the expected results. Due to the refraction and absorption of light in underwater environments, underwater images exhibit a wide range of variations and significant appearance discrepancy compared to terrestrial images (Raveendran et al., 2021; Zhang et al., 2022; Peng et al., 2023). Specifically, these discrepancy manifest in three main aspects: 1) **luminance**: As depth increases, light attenuates rapidly, resulting in gradually decreasing brightness. 2) **dynamic range**: Suspended particles in the water scatter light, causing the image's dynamic range to vary over a wide range in different water conditions. 3) **color cast**: Due to the different absorption rates of light wavelengths in water, longer wavelengths like red attenuate faster than shorter wavelengths like blue and green. This causes underwater images at greater depths to typically display a blue shift or green shift, while images in shallower areas maintain relatively normal colors. These significant variations in appearance are exactly why generative models pre-trained on terrestrial images (Rombach et al., 2022; Podell et al., 2023) perform poorly when generating underwater images.

To mitigate the negative impact of these appearance discrepancies, our research focuses on precise control of underwater image appearance. We model the appearance of underwater images using three parameters: luminance, dynamic range, and color cast, referred to as **appearance attributes**, and turn the appearance control of underwater images into a parameter-controlled conditional generation problem.

To address this issue, we propose a straightforward yet effective solution called SEADIFF. Our approach employs a Symmetrical Parameter Control framework to achieve precise control over the appearance of underwater images. This framework consists of two main modules: Appearance Writer (A-Writer) and Appearance Reader (A-Reader). 1) The **A-Writer** module encodes appearance attributes and utilizes a cross-attention mechanism (Vaswani et al., 2017) to inject these attributes into the U-Net encoder (Ronneberger et al., 2015). This enables the model to dynamically adjust for appearance discrepancies, significantly enhancing the visual consistency and realism of the generated underwater images. 2) The **A-Reader** module reads the feature maps from each layer of the U-Net decoder, predicting appearance attributes and providing deep supervision. This ensures that the appearance attributes predicted at different layers align with the expected appearance, maintaining high consistency between the generated images and the desired attributes. SEADIFF, through the combined use of these modules, effectively improves the controllability of underwater image appearance. As shown in Fig. 1, the proposed SEADIFF significantly enhances precise control over underwater image appearance while maintaining image quality and layout controllability.

In summary, the contributions of this paper are as follows:

1. We focused on the appearance discrepancies of underwater images and modeled them as three attributes: luminance, dynamic range, and color cast.

2. We proposed the Appearance Writer (A-Writer) module, which encodes appearance attributes and utilizes a cross-attention mechanism to inject these attributes into the U-Net encoder, significantly enhancing the visual consistency and realism of underwater images.

3. We proposed the Appearance Reader (A-Reader) module, which reads feature maps from each layer of the U-Net decoder, analyzes these features layer by layer to predict appearance attributes, and provides deep supervision to ensure that the generated images align with the expected appearance.

## 2 RELATED WORK

**Diffusion models.** Diffusion models have shown strong progress as generative models in recent years. Early models like DDPM (Ho et al., 2020) and DDIM (Song et al., 2020) generate clear images by progressively denoising Gaussian noise. This process transforms noise into high-quality images. Diffusion models are widely applied in tasks such as text-to-image synthesis (Nichol et al., 2021; Ramesh et al., 2022), image-to-image translation (Saharia et al., 2022a;b), image inpainting (Wang et al., 2022), and text-guided image editing (Nichol et al., 2021; Hertz et al., 2022). However, they struggle with underwater image generation due to challenges like light refraction and absorption.

**Layout-to-image generation** This area focuses on generating realistic images from layout information. Early methods like Layout2Im (Zhao et al., 2019) combined VAE (Kingma & Welling, 2013) and LSTM, using adversarial loss for realism and consistency. Recent approaches using GANs, such as LostGAN (Sun & Wu, 2019) and LAMA (Li et al., 2021), improve image quality. LAMA introduces a local perceptual mask adaptation module for handling overlapping object masks. Diffusion models have also made progress in this area. GLIGEN (Li et al., 2023) adds gated self-attention layers to a pre-trained diffusion model for better layout control, while Control-Net (Zhang et al., 2023) improves detail precision by incorporating semantic segmentation masks. ReCo (Yang Z & et al., 2023) uses both text and bounding boxes for text-to-image generation, and GeoDiffusion (Chen et al., 2023) supports geometric control with foreground prior re-weighting. However, these methods struggle with precise appearance control in underwater image generation, especially in terms of brightness, contrast, and color cast.

**Underwater image generation** Underwater image generation aims to create realistic underwater scenes. Early methods applied color correction techniques (Reinhard et al., 2001; Nguyen et al., 2014) to existing images. More recent work, such as GAN-based models (Liu et al., 2018), generates underwater images from terrestrial ones. (Desai et al., 2021) proposes a method that uses depth data to generate underwater images. Research using diffusion models (Zhang et al., 2024) has also explored generating realistic underwater images with terrestrial depth data. Additionally, (Desai et al., 2024) incorporates downwelling irradiance and direct light scattering for realistic synthetic underwater images. However, these methods still lack control over underwater image appearance.

## 3 PROBLEM STATEMENT

In this section, we first introduce the basic formulation of the problem discussed in Sec. 3.1. Then, we provide a detailed explanation of the three appearance attributes we modeled, including their computation methods in Sec. 3.2.

### 3.1 FORMULATION

The problem we focused on includes two aspects of control: layout control and appearance control.

As shown in Fig. 2, let $L = \{(c_i, b_i)_{i=1}^{N}\}$ be a layout with $N$ bounding boxes, where $c_i \in \mathcal{C}$ is the class of the bounding box and $b_i = [x_i, y_i, w_i, h_i]$ is the position and size of the bounding box in the image lattice ($H \times W$). Let $K_D = \{k_d\}_{d \in D}$ be the appearance attributes, where $D =$

Figure 2: **The pipeline for controlling appearance in underwater image generation.** Given the coarse information in the layout $L$ and the appearance attributes $K_D$, we train a model $\mathcal{G}$ to generate realistic underwater images $I_{\mathcal{G}}$.

$\{\text{luminance}, \text{dynamic range}, \text{color cast}\}$. The goal of the problem is to build a model $\mathcal{G}$, which can generate a realistic underwater image $I_{\mathcal{G}} \in \mathcal{R}^{3 \times H \times W}$, given the coarse information in the layout $L$ and the appearance attributes $K_D$.

## 3.2 APPEARANCE ATTRIBUTE

As discussed in the introduction, underwater images differ significantly from their terrestrial counterparts due to the unique effects of light refraction and absorption in aquatic environments. Specifically, these differences manifest in three primary aspects: luminance, dynamic range, and color cast. Understanding and controlling these attributes are crucial for generating realistic underwater images, as conventional models trained on terrestrial datasets often struggle to replicate the underwater appearance accurately. These challenges arise from the inherent discrepancies in appearance between terrestrial and underwater environments.

To address these challenges, we quantitatively model the three primary aspects as appearance attributes:

**Luminance.** Due to the rapid attenuation of light with increasing depth, underwater images exhibit significantly lower luminance compared to terrestrial images. To quantify this effect, we calculate luminance as follows: $k_L = \frac{1}{N} \sum_{i=1}^{N} I_i$, where $N$ is the total number of pixels, and $I_i$ is the grayscale value of the $i$-th pixel. This formulation captures the overall light intensity of the underwater image, reflecting decreased visibility at greater depths.

**Dynamic range.** In underwater environments, light scattering caused by suspended particles reduces dynamic range in images, making them appear flatter than terrestrial images. To quantify this effect, we calculate dynamic range as follows: $k_{DR} = \frac{1}{N} \sum_{i=1}^{N} (I_i - \bar{I})^2$, where $\bar{I}$ represents the average grayscale value of the image. This metric measures the variation in brightness, capturing the extent to which light scattering affects the dynamic range in underwater images.

**Color cast.** Due to the differential absorption of light wavelengths in water, longer wavelengths, such as red, attenuate more rapidly than shorter wavelengths like blue and green. This results in underwater images often appearing biased towards blue or green hues. We express color cast as follows: $k_{CC} = \frac{1}{N} \sum_{i=1}^{N} H_i$, where $H_i$ represents the hue value of the $i$-th pixel. This attribute describes how the colors in the image deviate from the actual scene colors, reflecting the color distortion caused by underwater conditions.

By accurately modeling these attributes, we transform the control of underwater image appearance into a parameter-based conditional generation problem. This approach allows us to effectively address discrepancies in appearance and generate more realistic underwater images.

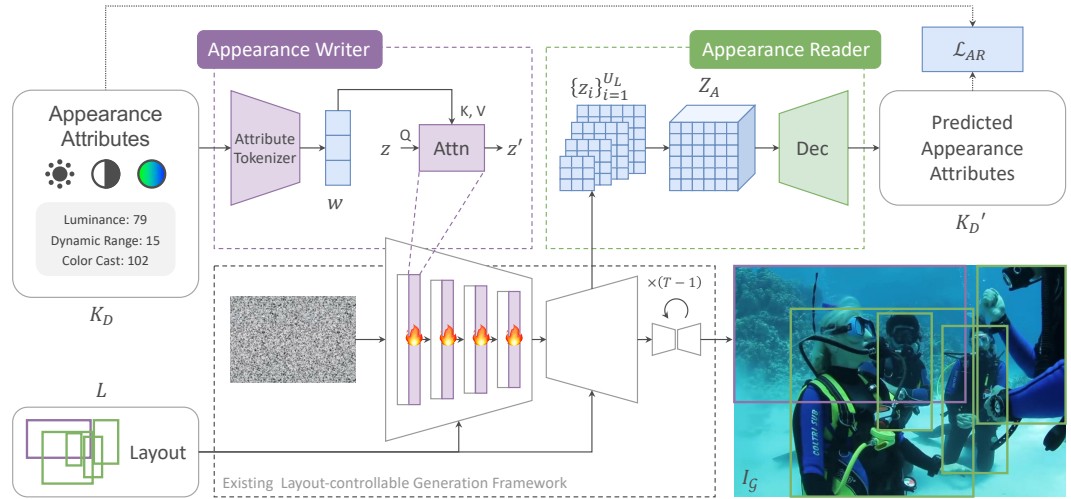

Figure 3: **Model architecture of our SEADIFF.** Our model extends the capability of layout-controllable generation by incorporating appearance attributes as input, allowing for simultaneous control over both layout and appearance. This model adds a Symmetrical Parameter Control component to an existing layout-controllable generation framework, which includes two modules: Appearance Writer (A-Writer) and Appearance Reader (A-Reader). A-Writer encodes the appearance attributes and injects them into the U-Net through Cross-Attention, while A-Reader reads the corresponding features from the U-Net and predicts the appearance attributes. A-Writer and A-Reader form a symmetric structure that jointly controls the appearance attributes.

## 4 SEADIFF

### 4.1 PRELIMINARY: CONDITIONAL DIFFUSION MODELS

Conditional diffusion models represent a significant advancement over traditional generative models such as GANs (Goodfellow et al., 2014) and VAEs (Kingma & Welling, 2013). These models learn the underlying data distribution through a $T$-step denoising process, beginning with normally distributed noise. This approach can be conceptualized as the inverse process of a fixed-length $T$ Markov chain (Ho et al., 2020).

In this framework, the model $\epsilon_\theta(x_t, t)$ is trained to reconstruct the clean data $x$ from a noisy input $x_t$ at a specific time step $t \in \{1, ..., T\}$ by estimating the noise added at that time. The training objective is defined as:

$$\mathcal{L}_{DM} = \mathbb{E}_{x, \epsilon \sim \mathcal{N}(0,1), t} \|\epsilon - \epsilon_\theta(x_t, t)\|^2. \tag{1}$$

An extension of this concept is found in Latent Diffusion Models (LDM) (Rombach et al., 2022), which perform the diffusion process in the latent space of a pre-trained Vector Quantized Variational AutoEncoder (VQ-VAE) (Van Den Oord et al., 2017). In LDM, the input image $x$ is first encoded into the VQ-VAE latent space, yielding $z = \mathcal{E}(x) \in \mathcal{R}^{H' \times W' \times D'}$. This latent representation serves as the clean sample in Eqn. 1.

To facilitate conditional generation, LDM introduces a conditional encoder $\tau_\theta(\cdot)$, modifying the objective to:

$$\mathcal{L}_{LDM} = \mathbb{E}_{\mathcal{E}(x), \epsilon \sim \mathcal{N}(0,1), t} \|\epsilon - \epsilon_\theta(z_t, t, \tau_\theta(y))\|^2, \tag{2}$$

where $y$ represents the condition applied, such as text in the context of LDM (Rombach et al., 2022).

### 4.2 METHOD OVERVIEW

The core of our proposed SEADIFF method lies in using Symmetrical Parameter Control to achieve precise control over underwater image appearance. It consists of two main modules: A-Writer and A-Reader. The A-Writer module (see Sec. 4.3) encodes appearance attributes and injects them into the U-Net via a cross-attention mechanism, allowing for fine control over the image features. Meanwhile, the A-Reader module (see Sec. 4.4) predicts appearance attributes by reading feature maps

from the U-Net decoder, providing deep supervision to ensure better alignment with the desired attributes and improving image quality. The combination of these modules enhances both appearance and layout control. For layout control, we use a grid-based method, where each grid position is represented by a unique position token encoded using the CLIP text encoder (Radford et al., 2021).

### 4.3 APPEARANCE WRITER

We utilize the Appearance Writer (A-Writer) module to regulate appearance in the diffusion model. In this work, we adopt a finely-tuned Vision Transformer (ViT) (Dosovitskiy et al., 2020) as the design for A-Writer to meet the control requirements for appearance. Experimental results in Sec. 5.4 demonstrate that this design effectively models appearance discrepancies.

As illustrated in Fig. 3, a shallow ViT is incorporated after each layer of the U-Net encoder to model these discrepancies. The appearance attributes $K_D$ are initially encoded into tokens using an attribute tokenizer:

$$w = \text{Tokenizer}(K_D). \tag{3}$$

Specifically, the attribute tokenizer is implemented as a shallow MLP that maps the appearance attributes $K_D \in \mathcal{R}^{|D| \times 1}$ into a vector representation $w \in \mathcal{R}^{|D| \times d_k}$, which serves as the token. Here, $|D|$ refers to the number of appearance attributes, and $d_k$ denotes the dimension of the keys in the cross-attention mechanism.

Next, the encoded tokens $w$ are injected into the U-Net through cross-attention (Vaswani et al., 2017):

$$z' = \text{Softmax}\left(\frac{QK^T}{\sqrt{d_k}}\right) V, \tag{4}$$

where $Q = W^Q z$, $K = W^K w$, and $V = W^V w$. The feature maps $z$ and $z'$ represent the U-Net feature maps before and after the update via the A-Writer module, respectively. The attention mechanism enables the diffusion model to take appearance attributes into account during the generation process.

During the fine-tuning process, we froze the U-Net encoder (Ronneberger et al., 2015) and trained only the proposed A-Writer along with the $K$ and $V$ parameters in the cross-attention mechanism. This approach enhances the model's ability to control the appearance of underwater images while preserving the quality of the generated results, enabling improved appearance control without compromising image fidelity.

### 4.4 APPEARANCE READER

The Appearance Reader (A-Reader) is another key component of our method. It aims to predict appearance attributes based on the feature maps extracted from each layer of the U-Net decoder. Inspired by (Luo et al., 2024), A-Reader analyzes feature maps at various levels to capture the intricate patterns and variations associated with appearance.

As shown in Fig. 3, A-Reader first extracts the corresponding feature maps $Z = \{z_i\}_{i=1}^{U_L}$ from each layer of the U-Net decoder, where each feature map $z_i$ has dimension of $\mathcal{R}^{H_i \times W_i \times C_i}$. Here, $H_i$, $W_i$, and $C_i$ represent the height, width, and number of channels, respectively, of the feature map at layer $i$, and $U_L$ denotes the total number of layers in the U-Net decoder.

Then, each feature map is resized to obtain standardized feature maps $\{z_i^s\}_{i=1}^{U_L}$ using the interpolation function, and these standardized feature maps are concatenated to form the aggregated feature map $Z_A$:

$$Z_A = \text{Concat}\left(\{\text{Interpolate}(z_i)\}_{i=1}^{U_L}\right). \tag{5}$$

After obtaining the aggregated feature map $Z_A$, we employ a decoder consisting of a Vision Transformer (ViT) (Dosovitskiy et al., 2020) and a multilayer perceptron (MLP) to predict the appearance attributes $K_D'$:

$$K_D' = \text{Dec}(Z_A) = \text{MLP}(\text{Self-Attn}(Z_A)). \tag{6}$$

The predicted appearance attributes $K_D' = \{k_d'\}_{d \in D}$ are then compared with the ground truth $K_D = \{k_d\}_{d \in D}$ to calculate the A-Reader loss $\mathcal{L}_{AR}$, obtained by computing the mean squared error (MSE)

between $K_D$ and $K'_D$:

$$\mathcal{L}_{AR} = \frac{1}{|D|} \sum_{d \in D} (k_d - k'_d)^2. \tag{7}$$

The final loss $\mathcal{L}$ is defined as the sum of the LDM (Rombach et al., 2022) loss $\mathcal{L}_{LDM}$ and the A-Reader loss $\mathcal{L}_{AR}$, expressed as $\mathcal{L} = \mathcal{L}_{LDM} + \mathcal{L}_{AR}$.

During the training process, A-Reader is jointly optimized with the rest of the model to ensure the consistency and accuracy of the predictions. The combination of A-Writer and A-Reader enables our method to control appearance in underwater images more effectively and accurately, providing better results in various applications.

# 5 EXPERIMENTS

## 5.1 IMPLEMENTATION DETAILS

**Dataset.** Our experiments were conducted on the RUOD dataset (Fu et al., 2023). To enhance the validity of our experiments, we annotated the appearance attributes introduced in Sec. 3.2 based on the RUOD dataset (Fu et al., 2023). However, the dataset contains multiple frames from the same video, leading to potential data leakage between training and testing sets due to the similarity of adjacent frames. To address this issue, we re-divided the RUOD dataset. Specifically, we used the first 10,000 images as the training set and the following 4,000 images as the testing set. Since the remaining 4,000 images still contained many similar images, we randomly selected 310 images from this subset for the final test set.

**Optimization.** We initialized our model using the pre-trained Stable Diffusion v1.5 model, which leverages the LDM (Rombach et al., 2022) text-to-image diffusion framework. All parameters in the U-Net (Ronneberger et al., 2015) were frozen, except for the $K$ and $V$ matrices. For training, we used the AdamW (Loshchilov & Hutter, 2019) optimizer, with a batch size of 32. The learning rate was set to $4 \times 10^{-5}$ for the U-Net and $3 \times 10^{-5}$ for the CLIP text encoder (Radford et al., 2021). All experiments were conducted on dual NVIDIA L40 GPUs, resulting in a total training time of 12 hours.

**Comparative experimental setting.** Given that our focus is on controlling both appearance and layout, and considering that most existing generative models do not support direct control over appearance, so we employ the layout-controllable generative models as comparative methods. We utilize their text interface to input appearance attributes. We construct the text prompts using the following template: "`brightness: $k_{\mathrm{L}}$, color cast: $k_{\mathrm{CC}}$, contrast: $k_{\mathrm{DR}}$.`" Furthermore, all training hyper-parameters are aligned with those specified in the original papers for the comparative methods. Furthermore, we do not include certain layout-controllable generative models that do not support text input, such as the classical work LayoutDiffusion (Zheng et al., 2023), in our comparisons.

**Evaluation metrics.** We use five metrics to evaluate the model's performance in terms of appearance controllability, generation quality, and layout controllability. For appearance controllability, we propose the Appearance MSE (A-MSE) metric, which measures the mean squared error (MSE) between the generated images and real images across the luminance, dynamic range, and color cast attributes. The A-MSE is defined as A-MSE $= \frac{1}{|D|} \sum_{d \in D} (k_d - k'_d)^2$. Generation quality is assessed using SSIM, PSNR , and FID (Heusel et al., 2017). Layout controllability is evaluated using the YOLO Score (Li et al., 2021). Specifically, we employ a YOLOv8 model (Reis et al., 2023b) that has been trained on the RUOD dataset (Fu et al., 2023) to compute the mean average precision at a 50% intersection over union (mAP@50) for the generated images.

## 5.2 QUALITATIVE RESULTS

In this section, we provide a qualitative evaluation of SEADIFF, comparing it with several existing methods on the RUOD dataset (Fu et al., 2023). We focus on three appearance attributes: luminance, dynamic range, and color cast. The comparison is visualized in three columns: the first shows the

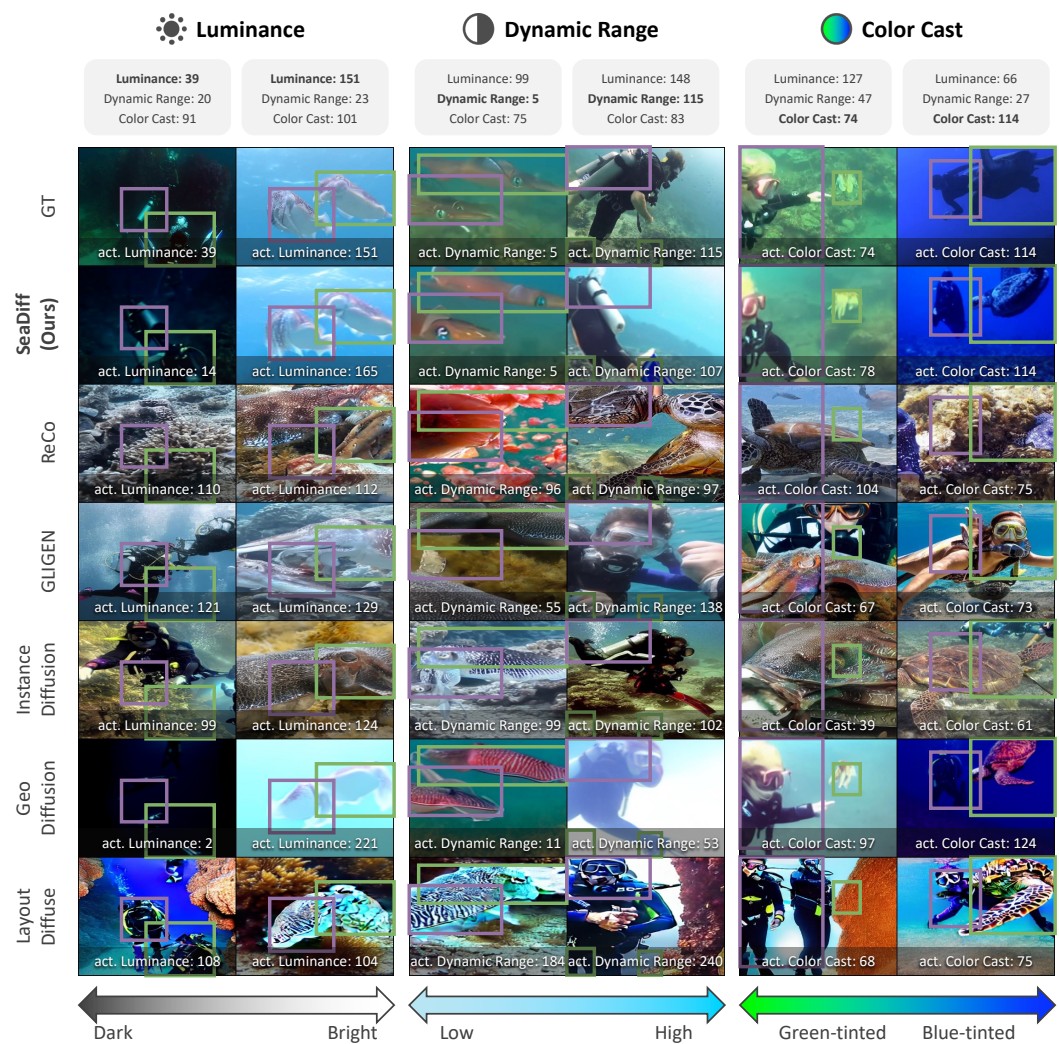

Figure 4: **Visualization of comparison with state-of-the-art methods on the RUOD dataset.** We present three representative cases of appearance attributes. The first column demonstrates the luminance attribute, ranging from dark to bright. The second column depicts the dynamic range attribute, transitioning from low to high. The third column illustrates the color cast attribute, varying from green-tinted to blue-tinted images. The label "act." on each image indicates the actual appearance attribute of the generated images. It can be observed that SEADIFF provides superior control over appearance while maintaining comparable image quality and layout consistency compared to existing methods.

variation in luminance (dark to bright), the second in dynamic range (low to high), and the third in color cast (green to blue).

Through a qualitative comparison with existing generative methods, we demonstrate the advantages of SEADIFF in underwater image generation tasks, as shown in Fig. 4. The results indicate that A-Reader generates images that are significantly closer to the GT images, particularly across the dimensions of luminance, dynamic range, and color cast. These findings suggest that SEADIFF offers more precise control over appearance attributes.

Unlike methods such as LayoutDiffuse (Cheng et al., 2023), which produce reasonably good object layouts but show noticeable deviations in luminance and dynamic range from the ground truth images, our method not only preserves an accurate object layout but also achieves more precise control over these attributes, resulting in generated images that closely resemble real underwater scenes.

Table 1: **Quantitative results on RUOD dataset.** The proposed SEADIFF demonstrates significant improvements across all evaluation metrics, showcasing enhanced appearance control, image quality, and layout consistency compared to previous work. "Input Res." refers to the input image resolution. *: represents the real image *Oracle* baseline.

| Method | Input Res. | A-MSE ↓ | SSIM↑ | PSNR↑ | FID↓ | YOLO Score↑ |
|---|---|---|---|---|---|---|
| *Oracle** | - | - | - | - | - | 0.768 |
| ReCo | 256×256 | 2047.4 | 0.086 | 10.602 | 73.263 | 0.005 |
| GLIGEN | 512×512 | 2122.2 | 0.132 | 10.490 | 69.161 | 0.020 |
| InstanceDiffusion | 512×512 | 2199.1 | 0.104 | 10.449 | 67.768 | 0.010 |
| GeoDiffusion | 256×256 | 1653.8 | 0.259 | 12.109 | 92.247 | 0.389 |
| LayoutDiffuse | 512×512 | 5169.1 | 0.107 | 9.248 | 84.395 | 0.274 |
| **SEADIFF (Ours)** | 256×256 | **375.1**$^{-1278.7}$ | **0.290**$^{+0.031}$ | **14.238**$^{+2.129}$ | **64.259**$^{-3.509}$ | **0.537**$^{+0.148}$ |

## 5.3 QUANTITATIVE RESULTS

**Quantitative results on RUOD dataset.** Table 1 presents a quantitative comparison of our proposed SEADIFF against existing methods on the RUOD dataset (Fu et al., 2023), showcasing significant improvements across all evaluation metrics, especially A-MSE. Specifically, our method achieved a minimum **A-MSE value of 375.1**, indicating outstanding appearance control capability, outperforming all baseline methods and reducing the appearance controllability metric by over 1200 compared to GeoDiffusion (Chen et al., 2023). Moreover, our method excels in image quality and spatial controllability, achieving the highest SSIM and PSNR values, as well as the best **FID score by reducing it by at least 3.509**, thus enhancing visual realism and diversity. Furthermore, our model attained the highest **YOLO Score of 0.537**, indicating that the integration of the appearance control module not only maintains but also enhances layout controllability.

**Zero-shot quantitative results on UTDAC dataset.** Table 2 presents the zero-shot performance of our proposed SEADIFF on the UTDAC dataset, compared to state-of-the-art methods. Notably, our approach exhibits outstanding generalization capabilities, consistently outperforming across all evaluation metrics without requiring any fine-tuning on the UTDAC dataset. Specifically, our method achieves a **A-MSE value of 83.7**, significantly exceeding all competing methods and showcasing its robust appearance control in unseen scenarios.

Overall, these results highlight the effectiveness of SEADIFF. The proposed A-Reader effectively enhances appearance controllability without compromising image quality or layout control. Quantitative results underscore the robustness and effectiveness of SEADIFF in achieving superior performance across multiple metrics.

Table 2: **Zero-shot Quantitative results on UTDAC dataset.** The proposed SEADIFF excels in the zero-shot setting, outperforming prior methods in appearance control, image quality, and layout consistency.

| Method | Input Res. | A-MSE ↓ | SSIM↑ | PSNR↑ | FID↓ |
|---|---|---|---|---|---|
| ReCo | 256×256 | 1149.3 | 0.173 | 14.199 | 149.677 |
| GLIGEN | 512×512 | 2105.1 | 0.182 | 12.579 | 196.895 |
| InstanceDiffusion | 512×512 | 2540.0 | 0.142 | 12.307 | 217.559 |
| GeoDiffusion | 256×256 | 658.8 | **0.617** | 16.903 | 144.576 |
| LayoutDiffuse | 512×512 | 2906.0 | 0.164 | 12.885 | 186.084 |
| **SEADIFF (Ours)** | 256×256 | **83.7**$^{-575.1}$ | 0.604 | **19.752**$^{+2.849}$ | **88.230**$^{-56.347}$ |

## 5.4 ABLATION STUDIES

To validate the contributions of A-Writer and A-Reader in our proposed method, we conducted a series of ablation studies, summarized in Table 3. Without any components, the appearance control capability was limited. Introducing A-Writer led to a significant decrease in the A-MSE, accompanied by improvements in SSIM and PSNR, indicating enhanced image quality. The use of A-Reader also yielded some improvements, though it was less effective in appearance control than A-Writer. Combining both components led to a further reduction in the **A-MSE to 375.1**, alongside improve-

Table 3: **Ablation study of the proposed modules A-Writer and A-Reader.** The table presents the quantitative evaluation metrics for different configurations, demonstrating the impact of each module on performance. "Param." refers to the number of model parameters, and "Inf. Time." denotes the average inference time per image.

| A-Writer | A-Reader | Param. | Inf. Time | A-MSE↓ | SSIM↑ | PSNR↑ | FID↓ | YOLO Score↑ |
|---|---|---|---|---|---|---|---|---|
| | | 1117M | 3805ms | 1359.2 | 0.269 | 12.910 | 76.589 | 0.451 |
| ✓ | | 1211M | 4455ms | 397.8 | 0.284 | 14.191 | 64.814 | 0.457 |
| | ✓ | 1124M | 3810ms | 490.2 | 0.270 | 13.994 | 67.740 | 0.462 |
| ✓ | ✓ | 1232M | 4846ms | **375.1**$^{-984.0}$ | **0.290**$^{+0.021}$ | **14.238**$^{+1.327}$ | **64.259**$^{-12.331}$ | **0.537**$^{+0.086}$ |

Table 4: **Downstream Task Results on the RUOD Dataset.** The table presents the performance of four representative object detection models trained on two datasets: RUOD and RUOD + SEADIFF. The results demonstrate that models trained on RUOD + SEADIFF consistently outperform those trained solely on RUOD, validating the practical value of SEADIFF-generated augmented data for improving downstream task performance.

| Data | YOLO | | Libra R-CNN | | Boosting R-CNN | | Faster R-CNN | |
|---|---|---|---|---|---|---|---|---|
| | mAP50 | mAP | mAP50 | mAP | mAP50 | mAP | mAP50 | mAP |
| RUOD | 0.567 | 0.421 | 0.698 | 0.492 | 0.764 | 0.541 | 0.692 | 0.468 |
| **RUOD + SEADIFF** | 0.668 | 0.502 | 0.771 | 0.525 | 0.781 | 0.579 | 0.718 | 0.470 |

ments in SSIM, PSNR, FID (Heusel et al., 2017), and YOLO Score (Li et al., 2021), highlighting a significant enhancement in performance.

These findings underscore the critical roles of A-Writer and A-Reader in enhancing appearance controllability, with their combination delivering the best overall performance. This emphasizes their complementary advantages in generating high-quality underwater images and precise appearance control.

## 5.5 DOWNSTREAM TASK

**Data augmentation.** To validate the practical value of the generated images, we used SEADIFF to generate an augmented dataset, RUOD + SEADIFF, based on the RUOD dataset. In the experiment, we used two sets of training data: one consisting of 10,000 real underwater images from the RUOD dataset, and the other combining 10,000 augmented images generated by SEADIFF with the RUOD dataset to form the RUOD + SEADIFF dataset. These augmented images exhibit rich variations in brightness, dynamic range, and color cast. We conducted experiments on four representative underwater object detection models as well as general object detection models, such as YOLOv8 (Reis et al., 2023a), Libra R-CNN (Pang et al., 2019), Boosting R-CNN (Song et al., 2023b), and Faster R-CNN (Ren et al., 2016). All models were trained on both training datasets and evaluated using the RUOD test set. The experimental results (see Table 4) show that models trained with the SEADIFF-augmented data (RUOD + SEADIFF) significantly outperform those trained only on the RUOD dataset in terms of performance on the RUOD test set. Visual results of the detectors on downstream tasks can be found in Appendix A.

## 6 CONCLUSION

This paper addresses the challenge of controlling underwater image appearance through three key attributes: luminance, dynamic range, and color cast. We introduce SEADIFF, a novel method that includes two main components: the Appearance Writer (A-Writer) and the Appearance Reader (A-Reader). Our experimental results demonstrate that SEADIFF enhances control over underwater image appearance, allowing precise adjustments in three key attributes while maintaining image quality and layout consistency.

**Limitations.** Although this study yields promising results, it only focuses on the appearance attributes of luminance, dynamic range, and color cast, without addressing the complex factors of underwater environments. Future research should explore how to model these factors for more precise control over underwater images.

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

APPENDIX

## A    MORE QUALITATIVE RESULTS

**Fixing the layout.** Since our method focuses primarily on controlling appearance, we fix the layout and generate images with different appearance attributes for the same layout. Specifically, we set luminance to $\{40, 64, 88, 112, 136, 160\}$, dynamic range to $\{10, 58, 106, 154, 202, 250\}$, and color cast to $\{40, 56, 72, 88, 104, 120\}$. These values are chosen because the image attributes in the dataset fall within these ranges, ensuring that the control aligns with real-world conditions. As demonstrated in Fig. 5, SEADIFF can precisely control image generation based on these varying appearance attributes, maintaining authenticity and consistency with real-world data.

**More generated images.** We visualize additional images generated by our SEADIFF method on the test set of RUOD dataset in Fig. 6, to further demonstrate its effectiveness in precisely controlling the appearance of underwater images. The generated images closely match the real images, maintaining spatial layout and ensuring high quality while achieving accurate control over appearance attributes. Upon observation, it is evident that the various attributes of the generated images are largely consistent with those of the real images, highlighting the robustness of our approach.

**Enhancing Downstream Tasks.** We conducted experiments on four representative object detection models, including the classic underwater detection model Boosting R-CNN, as well as the widely used general object detection models YOLO, Libra R-CNN, and Faster R-CNN. To validate the effect of the augmented data generated by SEADIFF on downstream tasks, we used two sets of training data: one consisting of 10,000 real underwater images from the RUOD dataset, and another combining the RUOD dataset with 10,000 augmented images generated by SEADIFF, forming the RUOD + SEADIFF dataset. The experimental results show that the models trained on RUOD + SEADIFF significantly outperform those trained on the RUOD dataset alone in terms of detection performance. As shown in Fig. 7, models trained with augmented data consistently show improved object detection accuracy, further validating the practical value and potential of images generated by SEADIFF for downstream tasks.

## B    MORE DISCUSSION

**General parameter control for image generation.** While our method primarily emphasizes controlling the appearance of underwater images, its core algorithms and control mechanisms can be effectively extended to parameter control in general image generation. This extension is of considerable practical significance, as the control parameters we employ—such as brightness, contrast, and color cast—are applicable across a wide range of image types. By leveraging cross-attention mechanisms and deep supervision techniques, our approach achieves precise control over the appearance of underwater images, and these techniques hold great potential for application in various other domains.

## C    MORE APPLICATION

**Data generation.** Our model possesses the capability to exert a broad range of control over appearance, making it an effective tool for data generation. In the field of underwater image processing, existing datasets often suffer from insufficient samples and a lack of diversity. Our model addresses these shortcomings by generating high-quality underwater images. Specifically, the model simulates diverse underwater environments by adjusting various generation parameters, such as lighting conditions, color distribution, and object characteristics. This flexibility not only enhances the realism of the generated images but also encompasses a wide range of potential underwater scenarios, enriching the existing dataset. The generated images can be utilized for training and validating underwater algorithms, improving their adaptability to different environmental changes, thereby enhancing algorithm performance.

**Enhancing Existing Data.** Real underwater datasets are often composed of continuous video footage collected from the same marine area, resulting in relatively low diversity and minimal appearance variation in the images. Consequently, these datasets can only represent specific environ-

mental characteristics, limiting the robustness of model training. Given that our model has been trained on a large-scale appearance dataset, it possesses the ability to generate diverse appearances. We can fine-tune this model on real datasets with lower diversity, enabling it to produce images similar to those in these datasets while retaining control over a wide range of appearance variations. In this way, we can generate numerous images that resemble the original dataset, while introducing diverse appearances, effectively augmenting the existing dataset and enhancing its utility in deep learning model training.

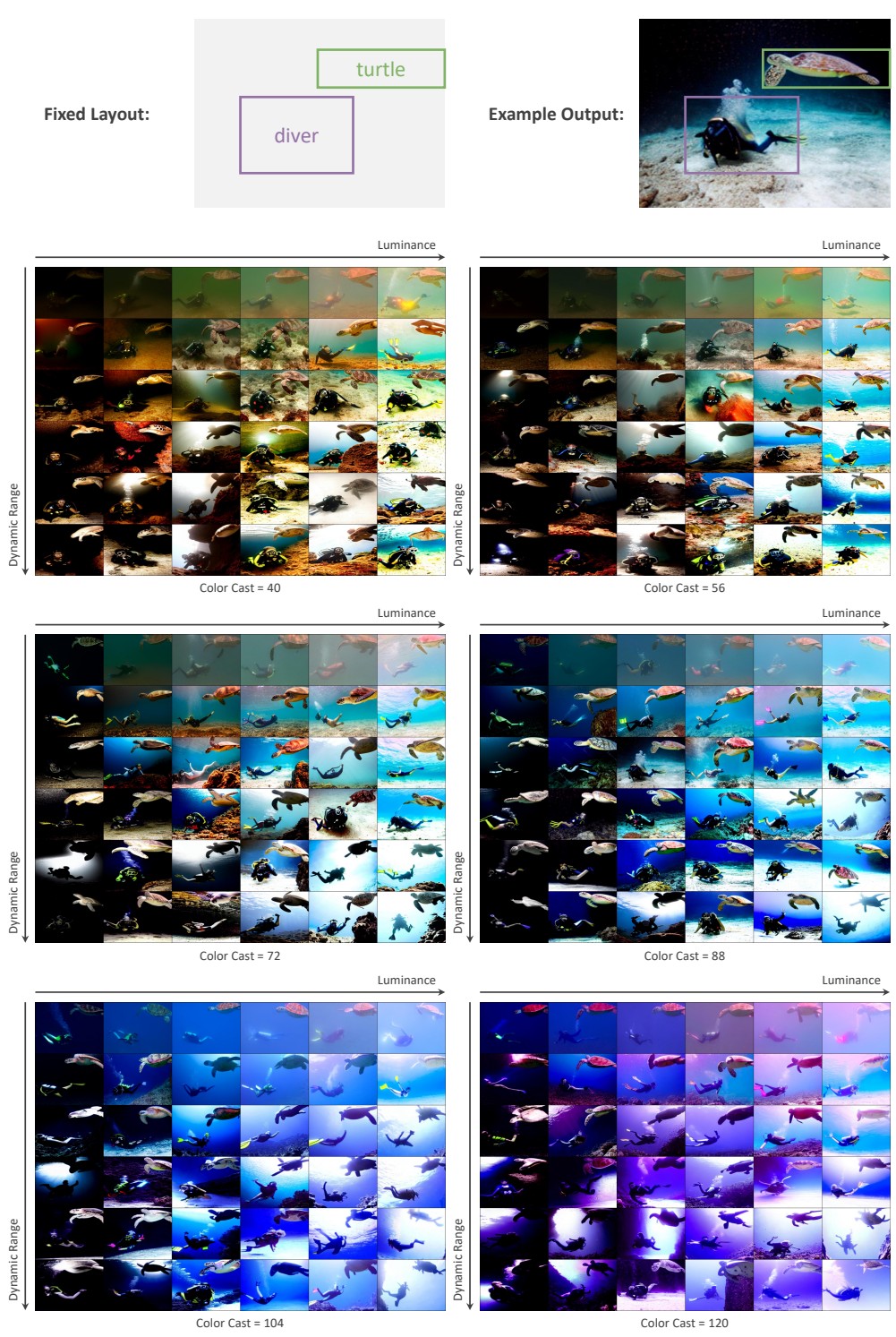

Figure 5: **Visualization of changing appearance attributes while fixing the layout.** For the top two images, the left one is the input layout and the right one is the example output. For the following six images, they correspond to different color cast attributes that increase sequentially. For each sub-image, the horizontal coordinate is luminance and the vertical coordinate is dynamic range. It can be seen that the proposed SEADIFF can precisely control the appearance of underwater images within a certain range.

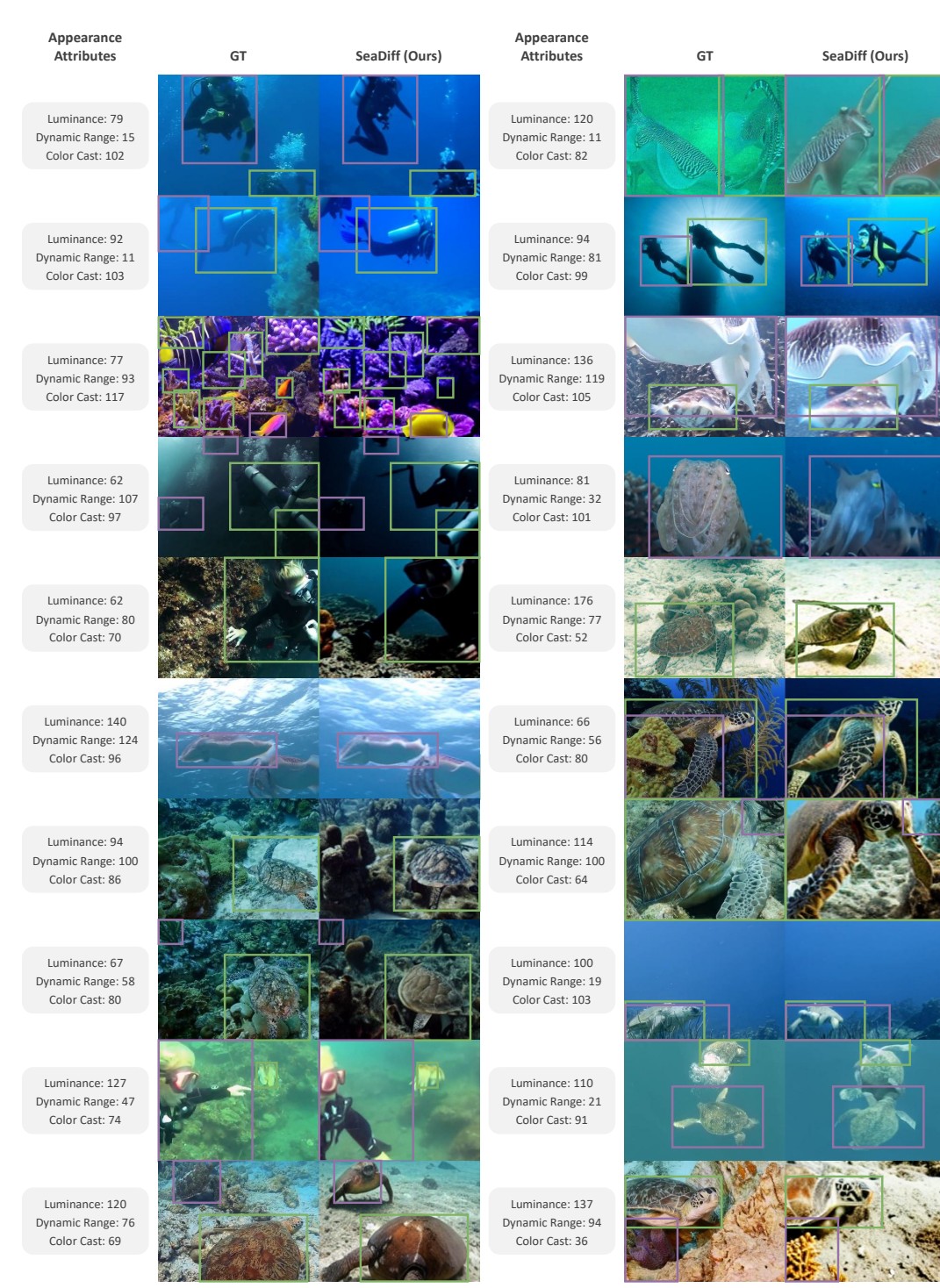

Figure 6: **More results on the test set of the RUOD dataset.** This figure showcases additional images generated by our SEADIFF method. The first column displays the appearance attributes, while the second and third columns show the ground truth (GT) images and generated images, respectively. Each generated image closely resembles its corresponding real image, illustrating the method's effectiveness in controlling the appearance of underwater images. The generated images maintain layout and high quality, with attributes that largely align with the real images, underscoring the robustness of our approach.

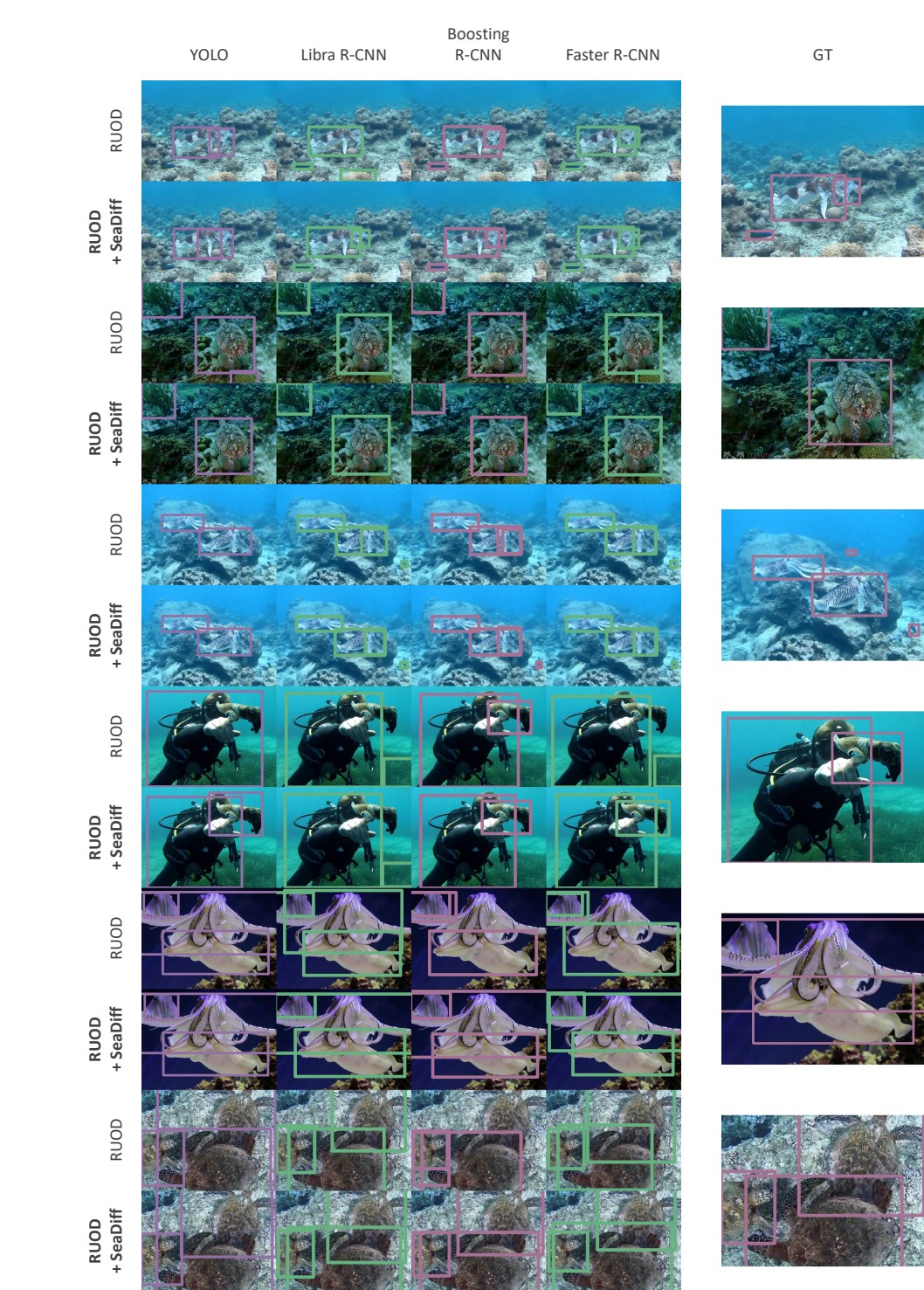

Figure 7: **The benefits of SEADIFF data augmentation for underwater object detection tasks.** By comparing the original training set RUOD with the augmented dataset (RUOD + SEADIFF), it can be observed that models trained on the augmented dataset achieve superior performance in underwater object detection tasks. This indicates that SEADIFF, as a data augmentation method, not only effectively improves data quality but also plays a positive role in enhancing the performance of downstream tasks.

