# OpenReview forum: "SeaDiff: Delve into Underwater Image Generation with Symmetrical Parameter Control"
_ICLR.cc/2025/Conference — Submitted to ICLR 2025_

### Official Review · Reviewer_PBAu · 2024-10-27

**Soundness:** 3
**Presentation:** 3
**Contribution:** 2
**Rating:** 3
**Confidence:** 4

**Summary:**

This paper proposed a new underwater data generation method, which can control the luminance, dynamic range, and color cast of the generated images. Experiments prove that this method achieves superior control over appearance while maintaining image quality and layout-consistency. However, its application is limited and its techniques have been widely studyed. Specific reasons are given in weakness.

**Strengths:**

1) This article achieves precise control over underwater image generation, and the performance is good.
2) To some extent, I believe that this method can serve as an alternative to underwater image enhancement, but the paper itself did not discuss this point.
3) This paper introduces the simultaneous use of luminance, dynamic range, and layout control for underwater image generation.

**Weaknesses:**

1) its application is limited. From my opinion, the underwater image enhancement and some downstream applications, e.g., object detection and semantic segmentation, are significant. Only generating underwater images is meaningless but this paper only discusses how the proposed method can effectively and controllably generate underwater images, but does not mention the significance of generating underwater images. Only generating images of a specific scene is meaningless.

2) Besides, the techniques it used have been widely adopted. If this paper proposed a novel technique for underwater image generation, it will still be a valuable work. However, the A-Writer and A-Reader are not novel and interesting enough. Their key contributions are cross-attention and feature supervision. The layout control is also achieved by cross-attention. These techniques have been widely used and are not designed for underwater image generation.

This paper lacks application impact and technological innovation.

**Questions:**

1) The impact on mainstream applications should be fully discussed, including image enhancement and downstream applications. Generating underwater images only is not important.

2) Underwater-oriented techniques should be studied. Considering the content of this task, I believe it would be more meaningful to design specialized generative components for underwater scenes.

---

> ### Author Response · Authors · 2024-11-21
>
> ### Weekness 1: lacking applications
>
> Thank you for the reviewer’s comments. We understand the concern regarding the question of whether generating underwater images alone is meaningful. We believe that the core value of generating underwater images lies in their practical application in downstream tasks such as object detection and semantic segmentation. As mentioned in the paper, existing underwater datasets suffer from significant limitations in appearance diversity, which directly impacts the performance of downstream tasks, particularly in terms of robustness to different underwater environments. With our SeaDiff method, we can generate high-quality underwater images with precise appearance control, greatly enhancing data augmentation in scenarios with insufficient diversity.
>
> In the experimental section, we have preliminarily demonstrated the application value of SeaDiff in downstream tasks. By using the generated diverse underwater images to augment existing datasets, we observed a significant improvement in the accuracy of a YOLO-based object detection model on the RUOD dataset. This shows that SeaDiff not only generates underwater images with precise appearance control but also significantly boosts downstream task performance through data augmentation, fully demonstrating the practical value of generated images.
>
> To further validate our approach, we conducted additional experiments by evaluating four representative object detection models. The detailed experimental results can be found in Question 1, and we believe these results will further demonstrate the effectiveness and potential of SeaDiff in downstream tasks.
>
> ### Weekness 2: lacking novelty and insight
>
> We appreciate the reviewer’s perspective on the need for underwater image generation to align closely with real-world applications. The proposed method not only enables precise generation of underwater images with specific attributes but also serves as a powerful data augmentation tool. By enriching training datasets for underwater perception tasks such as object detection and semantic segmentation, it effectively improves performance in downstream applications. Additionally, by generating low-quality underwater images that mimic real-world degradation characteristics, the method offers more diverse training data, enhancing the robustness of underwater perception systems.
>
> We understand the reviewer’s concerns about the novelty of the techniques. While A-Writer and A-Reader leverage cross-attention mechanisms and feature supervision, which have been explored in other fields, the innovation of this work lies in adapting and customizing these techniques specifically for underwater image generation. The model introduces a novel approach to precise control of underwater image attributes, such as luminance, dynamic range, and color cast, which is a new attempt in this domain. By tailoring these mechanisms to address the unique challenges of underwater scenarios, this work provides a fresh perspective and demonstrates the potential for advancing underwater image generation methods.

---

> ### Author Response · Authors · 2024-11-21
>
> ### Question 1: impact on mainstream applications
>
> SeaDiff, with its precise control over image appearance attributes, not only provides an innovative solution for image enhancement but also offers high-quality data to support downstream applications, driving advancements in underwater-related research and development.
>
> To evaluate the practical value of generated images, we conducted additional experiments using four representative object detection models:
> - **Boosting R-CNN**: A classic underwater object detection model.
> - **YOLO**, **Libra R-CNN**, and **Faster R-CNN**: Widely used general-purpose object detection models.
>
> The experiments involved two sets of training data:
> 1. **RUOD**: Consisting of 10,000 real underwater images.
> 2. **RUOD_Aug**: A combination of the RUOD dataset (10,000 images) and 10,000 augmented images generated using SeaDiff, which exhibit diverse luminance, dynamic range, and color cast characteristics.
>
> All models were trained on these two datasets and evaluated using the RUOD test set. The results are as follows:
>
> | Data     | YOLO  | YOLO  | Libra-RCNN | Libra-RCNN | Boosting-RCNN | Boosting-RCNN | Faster-RCNN | Faster-RCNN |
> |------------|-------|-------|------------|------------|---------------|---------------|-------------|-------------|
> |            | mAP50 | mAP50-95 | mAP50     | mAP50-95   | mAP50         | mAP50-95      | mAP50       | mAP50-95    |
> | RUOD       | 0.567 | 0.421 | 0.698      | 0.492      | 0.764         | 0.541         | 0.692       | 0.468       |
> | **RUOD_Aug** | **0.668** | **0.502** | **0.771**  | **0.525**  | **0.781**     | **0.579**     | **0.718**   | **0.470**   |
>
> The results demonstrate that whether it is the Boosting R-CNN designed specifically for underwater scenarios or general object detection models (YOLO, Libra R-CNN, Faster R-CNN), the use of augmented data generated by SeaDiff significantly improves model performance. This experiment clearly validates the practical value and immense potential of generated images in downstream applications.
>
> ### Question 2: underwater-oriented techniques should be studied
>
> Designing dedicated underwater generation components can optimize specific underwater attributes, but it may also limit the applicability of the model to other scenarios. While underwater environments have unique characteristics, such as low illumination, significant dynamic range variations, and color shifts, their complexity and diversity mean that a single specialized component might struggle to comprehensively address all possible cases. Moreover, developing and training specialized components require a large amount of specific data, which is particularly challenging given the inherent scarcity of underwater datasets, further increasing development difficulty and costs.
>
> In contrast, SeaDiff adopts a more general and flexible design approach. By introducing a symmetrical parameter control framework, it achieves precise parametric control over key underwater image attributes, such as luminance, dynamic range, and color cast. This design not only enhances the visual quality and consistency of generated images but also allows the model to adapt seamlessly to various underwater environments, avoiding the need to create tailored solutions for each specific scenario.

---

### Official Review · Reviewer_HkDY · 2024-10-30

**Soundness:** 2
**Presentation:** 3
**Contribution:** 2
**Rating:** 5
**Confidence:** 4

**Summary:**

This work models the appearance of underwater images from three aspects: luminance, dynamic range, and color cast,  achieving precise control over the appearance of underwater images. Additionally, this paper proposes a new method, SEADIFF, which comprises two primary contributions: the Appearance Writer, responsible for encoding and injecting appearance attributes into the U-Net encoder; and the Appearance Reader, which ensures that the generated images align with the desired appearance by analyzing the feature maps.

**Strengths:**

1.This work models the appearance of underwater images across three aspects: luminance, dynamic range, and color cast. The idea is interesting.

2.This paper is written with clarity and is highly accessible to readers.

**Weaknesses:**

1. The proposed method is relatively simple, lacking novelty and insight. This work appears to be more of an engineering effort rather than an academic innovation.

2. Although the proposed method achieves superior results based on quantitative metrics, the visualizations make it difficult to discern its advantages. For instance, in Figure 1, it is challenging to observe that the images generated by our method exhibit higher quality compared to those produced by GeoDiffusion.

3.The experiments are not robust, and additional studies are needed to demonstrate that the images generated by the proposed method are beneficial.

**Questions:**

1. Why do the input resolutions differ across the various methods presented in Table 1?

---

> ### Author Response · Authors · 2024-11-23
>
> ### Weekness 1: lacking novelty and insight
>
> We sincerely thank the reviewers for their valuable feedback. We recognize the need to further elucidate the innovation in our module design and would like to clarify the contributions of our research through the following points.
>
> First, the primary focus of our study is not merely the development of new modules but addressing a long-standing and critical challenge in underwater image generation: achieving precise control over image appearance (luminance, dynamic range, and color cast) under highly challenging optical conditions. While A-Writer and A-Reader are relatively straightforward in their technical implementation, their design is fully aligned with our goal-driven research. Through the innovative symmetrical parameter control framework, these modules effectively address the uncertainties and complexities of underwater image appearance. The core value of this framework lies in the unique integration of A-Writer’s appearance encoding and injection mechanism with A-Reader’s layered supervision mechanism, enabling high-precision alignment between generated images and specified appearance attributes.
>
> Second, the rationale and novelty of the module selection are strongly supported by quantitative experiments. Ablation study results (Section 5.4, Table 2) show that both A-Writer and A-Reader significantly enhance appearance control performance individually, and their combined effect further reduces A-MSE from 1359.2 (without modules) to 375.1. Additionally, other metrics, such as SSIM, PSNR, and FID, also exhibit substantial improvement. These results demonstrate that our module design is not arbitrary but grounded in a deep understanding of the problem and extensive experimental validation. This design enables SeaDiff to surpass existing underwater image generation methods across multiple key metrics, as shown in Table 1 and Figure 4.
>
> Furthermore, the combination of A-Writer and A-Reader is not merely an engineering solution but represents a novel approach to modularly controlling the behavior of diffusion models under complex conditions. Existing methods, such as GeoDiffusion and LayoutDiffuse, primarily focus on geometric control or layout consistency but exhibit significant shortcomings in appearance control. Our work bridges this gap through the symmetrical collaboration between the two modules, achieving comprehensive control over key appearance attributes in underwater environments for the first time.
>
> Lastly, it is important to emphasize that the significance of our method extends beyond the specific application of underwater image generation. The core mechanisms of A-Writer and A-Reader, such as cross-attention and the integration of deep supervision, possess strong generalizability and can be applied to other generative tasks requiring appearance control. We believe this approach provides a new perspective for future research on image generation models and holds significant academic value and practical potential.
>
> ### Weekness 2: visualizations make it difficult to discern its advantages
>
> The core contribution of this work lies in achieving precise control over underwater image attributes, rather than merely pursuing higher image quality. In Figure 1, the advantage of the SeaDiff method is primarily reflected in its precise control of attributes such as luminance. For instance, as the luminance control parameter increases, the GeoDiffusion method tends to enhance the overall brightness of the image, resulting in overexposure and loss of detail. In contrast, SeaDiff can reasonably allocate brightness while increasing luminance, ensuring that the image retains a natural visual appearance. This ability to control attributes highlights the unique value of our approach.
>
> Furthermore, the primary goal of this study is to generate underwater images with varying degradation characteristics through attribute control to augment underwater datasets and improve the performance of underwater perception algorithms. Therefore, it is unfair to evaluate our method solely based on image quality, as our focus is on the controllability and usability of the generated image attributes.
>
> In addition, SeaDiff significantly enhances the accuracy of attribute control while maintaining image quality. As shown in Table 1, compared with other methods, SeaDiff achieves slight optimization in the FID metric and a substantial decrease in the A-MSE metric, indicating a certain improvement in image quality and a notable enhancement in attribute controllability. This balance between quality and control capability demonstrates the higher practical value of SeaDiff in real-world applications.

---

> ### Author Response · Authors · 2024-11-23
>
> ### Weekness 3: additional studies to validate the benefits of the generated images
>
> The proposed SeaDiff method is a generative approach that enables precise control over the appearance of underwater images. By generating images with diverse visual characteristics, it enhances existing datasets, thereby improving the performance of downstream tasks.
>
> As shown in the comparative experiments in the article (see Table 1), the YOLO detector trained on the RUOD dataset achieved an mAP of 0.768 on the RUOD test set using ground truth (GT) images. When trained on data generated by SeaDiff, the YOLO detector achieved an mAP of 0.537. Although slightly lower than the GT images, the performance is comparable, demonstrating the potential of generated images in detection tasks. Moreover, the generated images exhibit a greater variety of appearance variations. These initial results highlight the practical utility of the generated images.
>
> To further validate the value of generated images in real-world applications, we conducted additional experiments using four representative object detection models:
> - **Boosting R-CNN**: A classic underwater object detection model.
> - **YOLO**, **Libra R-CNN**, and **Faster R-CNN**: Classic general-purpose object detection models.
>
> The experiments utilized two training datasets:
> 1. **RUOD**: A dataset consisting of 10,000 real underwater images.
> 2. **RUOD_Aug**: A dataset combining the RUOD dataset (10,000 images) with 10,000 augmented images generated using SeaDiff, which feature diverse brightness, dynamic range, and color cast variations.
>
> All models were trained on these two datasets and evaluated on the RUOD test set. The experimental results are as follows:
>
> | Data     | YOLO  | YOLO  | Libra-RCNN | Libra-RCNN | Boosting-RCNN | Boosting-RCNN | Faster-RCNN | Faster-RCNN |
> |------------|-------|-------|------------|------------|---------------|---------------|-------------|-------------|
> |            | mAP50 | mAP50-95 | mAP50     | mAP50-95   | mAP50         | mAP50-95      | mAP50       | mAP50-95    |
> | RUOD       | 0.567 | 0.421 | 0.698      | 0.492      | 0.764         | 0.541         | 0.692       | 0.468       |
> | **RUOD_Aug** | **0.668** | **0.502** | **0.771**  | **0.525**  | **0.781**     | **0.579**     | **0.718**   | **0.470**   |
>
> The results indicate that models trained on augmented data (RUOD_Aug) generated by SeaDiff outperformed those trained on the original dataset across all metrics:
> - **Boosting R-CNN** showed improvements of 1.7% in mAP50 and 3.8% in mAP50-95.
> - **YOLO** demonstrated significant gains, with mAP50 and mAP50-95 increasing by 10.1% and 8.1%, respectively.
> - **Libra R-CNN** improved by 7.3% in mAP50 and 3.3% in mAP50-95.
> - **Faster R-CNN** achieved enhancements of 2.6% in mAP50 and 0.2% in mAP50-95.
>
> In summary, whether for underwater-specific models like Boosting R-CNN or general object detection models such as YOLO, Libra R-CNN, and Faster R-CNN, the augmented data generated by SeaDiff significantly enhances model performance. This further validates the practical value and potential of using generated images in downstream applications.
>
> ###  Question 1:
>
> The differences in input resolutions across the methods presented in Table 1 primarily stem from their use of default resolution settings, which are typically optimized and selected based on the architectural design characteristics and specific task requirements of each method. In this study, we retained all models' default configurations and minimized any unnecessary modifications to ensure that each method performs optimally within its original design framework. This approach maximizes the demonstration of each method's true effectiveness while ensuring the fairness of the comparative experiments.

---

### Official Review · Reviewer_Cfjp · 2024-10-30

**Soundness:** 2
**Presentation:** 2
**Contribution:** 2
**Rating:** 3
**Confidence:** 5

**Summary:**

This paper tackles the challenge of managing underwater image appearance by focusing on three key attributes: illumination, dynamic range, and color cast. The method comprises two primary components: Appearance Writer (A-Writer) and Appearance Reader (A-Reader). The former encodes and injects appearance attributes into the U-Net encoder, while the latter ensures that the generated image aligns with the desired appearance by analyzing the feature maps.

**Strengths:**

This paper is structured in a clear and accessible manner, making it easy to follow for readers. The topic of generating diverse and realistic underwater images is particularly intriguing, as it addresses the challenges faced in underwater imaging and highlights the potential for advancements in this field.

**Weaknesses:**

1. The results presented are not satisfactory. As shown in Fig. 1, the images fail to meet the expected standards of realism.
2. Why is it important to generate underwater images, particularly those of low quality (severely degraded)? Will this be beneficial for downstream tasks, or does it serve other purposes?
3. The author creates new underwater images based on three attributes: illumination, dynamic range, and color cast. However, the degree of turbid, a key attribute of underwater images is neglected.

**Questions:**

1. Can the proposed method generate high-resolution underwater images?
2. What is the inference efficiency of the proposed method?

---

> ### Author Response · Authors · 2024-11-21
>
> ### Weekness 1: fail to meet the expected standards of realism
>
> 1. **Theoretical Analysis: Balancing Model Complexity and Image Quality**
>     Thank you for your valuable feedback. From a theoretical perspective, introducing additional control variables in generative models inevitably increases system complexity. This complexity may affect the quality of generated images in high-dimensional spaces. Therefore, finding an optimal balance between control capabilities and image quality remains a key challenge in the field of generative modeling.
> 2. **Methodology Focus: Enhancing Appearance Control While Preserving Quality**
>     Our research goal is to ensure image quality while significantly improving control over appearance. To this end, we designed the A-Writer and A-Reader modules, which substantially optimize the model's control over appearance. During our experiments, we observed that simply increasing control variables could lead to a decline in image quality. To address this, we adopted a strategy of freezing the pre-trained parameters of U-Net and fine-tuning only a subset of parameters. This approach minimizes potential quality loss while significantly enhancing control capabilities. We believe that in practical applications, the precision and consistency of appearance control are crucial for user experience. Thus, our work focuses not only on achieving visual realism but also on advancing the model’s ability to control appearance effectively.
> 3. **Experimental Validation: Evidence from Table 1**
>     As shown in the results presented in Table 1, our method achieves an excellent balance between image quality and appearance control. Compared to baseline models, our method not only maintains stability in image quality scores but also achieves a slight improvement. Moreover, it significantly enhances the appearance control metric (A-MSE). These results demonstrate that our design successfully enhances appearance controllability while preserving image quality, validating both the effectiveness and innovation of our approach.
>
> ### Weekness 2: why is it important to generate low quality underwater images
>
> Generating low-quality underwater images plays a significant role in improving downstream task performance, achieving efficient data augmentation, and advancing research in the field of underwater image enhancement. These images realistically simulate underwater degradation characteristics such as insufficient lighting, color distortion, and low contrast, significantly enriching the diversity of datasets. This diversity enables models to better adapt to complex underwater environments, thereby enhancing their robustness in tasks like object detection and semantic segmentation. Our research is the first to propose a method for controllable generation of underwater image appearances, effectively filling a research gap in this field.
>
> To validate the utility of generating low-quality underwater images for downstream tasks, we conducted further experiments. Four classic object detection models (Boosting R-CNN, YOLO, Libra R-CNN, and Faster R-CNN) were trained on two datasets: the real underwater image dataset (RUOD) and a mixed dataset (RUOD_Aug) enhanced with generated data. The models were evaluated on the RUOD test set. The results showed significant performance improvements when augmented data generated by SeaDiff was included: Boosting R-CNN achieved an mAP50 and mAP50-95 improvement of 1.7% and 3.8%, respectively; YOLO showed the most notable improvement with mAP50 and mAP50-95 increasing by 10.1% and 8.1%, respectively; Libra R-CNN and Faster R-CNN saw mAP50 improvements of 7.3% and 2.6%, and mAP50-95 improvements of 3.3% and 0.2%, respectively. These findings strongly validate the potential and value of generated data in enhancing model performance and practical applications.
>
> | Data     | YOLO  | YOLO  | Libra-RCNN | Libra-RCNN | Boosting-RCNN | Boosting-RCNN | Faster-RCNN | Faster-RCNN |
> |------------|-------|-------|------------|------------|---------------|---------------|-------------|-------------|
> |            | mAP50 | mAP50-95 | mAP50     | mAP50-95   | mAP50         | mAP50-95      | mAP50       | mAP50-95    |
> | RUOD       | 0.567 | 0.421 | 0.698      | 0.492      | 0.764         | 0.541         | 0.692       | 0.468       |
> | **RUOD_Aug** | **0.668** | **0.502** | **0.771**  | **0.525**  | **0.781**     | **0.579**     | **0.718**   | **0.470**   |

---

> ### Author Response · Authors · 2024-11-21
>
> ### Weekness 3: degree of turbid is neglected
>
> We fully acknowledge that turbidity is indeed a critical environmental attribute of underwater images. In this study, our primary objective is to propose an attribute-controlled image generation method, using brightness, dynamic range, and color cast as examples for initial exploration. These attributes were chosen based on their universality in underwater environments and their ease of quantification and control, which makes them suitable for verifying the feasibility and effectiveness of our method.
>
> It is worth emphasizing that **our SeaDiff framework is highly versatile and extensible**. While the current experiments do not yet control for attributes like turbidity, our framework can be readily adapted by adjusting the attribute injection and supervision modules in A-Writer and A-Reader to accommodate other environmental attributes, including turbidity. The design of these modules inherently supports diverse attribute extensions and is not limited to the attributes selected in the current experiments.
>
> In the future, we plan to incorporate critical underwater attributes such as turbidity, visibility, and lighting uniformity into our research. By specifically improving A-Writer and A-Reader to enhance attribute representation and supervision, we aim to further boost the performance of SeaDiff in generating complex underwater images. This will also significantly expand its applicability and robustness in extreme scenarios and practical applications.
>
> In summary, although turbidity is not included in the current experiments, our framework lays a solid foundation for future extensions. We believe that by introducing more environmental attributes and optimizing the generation mechanism, the potential of SeaDiff can be further unlocked, leading to breakthroughs in generating complex underwater images.
>
> ### Question 1: generate high-resolution underwater images
>
> Our method is a resolution-agnostic framework, with its core focus on attribute-controlled underwater image generation. It is worth emphasizing that existing diffusion models inherently possess the capability to generate high-resolution images, making our method easily adaptable to these high-resolution models for more detailed image generation.
>
> ### Question 2: inference efficiency
>
> We fully understand the importance of inference efficiency, including model parameter size and inference latency, in evaluating the practicality and performance of methods. Although this aspect was regrettably overlooked in our initial experiments, we have subsequently conducted additional tests, and the results are presented in the table below, showing the model parameter sizes and inference latencies for different configurations of the A-Writer and A-Reader modules. All inference latency data were measured on a single NVIDIA RTX 3090 with the batch size set to 1 during testing. It should be noted that this setup does not fully reflect actual inference efficiency, as batch sizes in real-world applications are typically larger, enabling faster single-image inference. The batch size was set to 1 here solely for the purpose of simplifying and standardizing the evaluation process.
>
> | **A-Writer** | **A-Reader** | **Param** | **Inf. Time** |
> | ------------ | ------------ | --------- | ------------- |
> |              |              | 1117M     | 3805ms        |
> | ✓            |              | 1211M     | 4455ms        |
> |              | ✓            | 1124M     | 3810ms        |
> | ✓            | ✓            | 1232M     | 4846ms        |
>
> The results show that the A-Writer module significantly improves generative capabilities while only adding 94M parameters and 650ms of inference time, demonstrating a reasonable overhead. Meanwhile, the A-Reader module enhances reading comprehension capabilities with minimal cost, adding only 7M parameters and 5ms of inference time. When the two modules are combined, the total overhead amounts to 115M parameters and 1041ms of inference time, achieving a good balance between performance improvement and inference efficiency. This design effectively integrates functionality and efficiency, making it suitable for diverse real-world application scenarios.

---

> > ### Comment · Reviewer_Cfjp · 2024-11-28
> >
> > Thanks for your detailed response. Generating realistic underwater images is an interesting topic. However, the rebuttal regarding the quality of generated images and their applications do not convince me. First, there is a noticeable gap between the generated images and real-world underwater images. Second, the performance on downstream tasks requires further discussion, such as other applications and comparison to other methods. Based on the responses and the comments from other reviewers, I maintain my original score.

---

> > > ### Author Response · Authors · 2024-12-01
> > >
> > > Thank you for your response, but we do not agree with your viewpoint.
> > >
> > > Firstly, you mentioned that there is a significant gap between the generated images and real underwater images. We would like to emphasize that the focus of this paper is on controlling the attributes of underwater images. The images generated by our method are more realistic in terms of these attributes. Additionally, our method outperforms others in quantitative metrics such as FID and YOLO Score, which largely reflects the sensory and layout similarity between the generated images and real images, further proving the realism of the generated images.
> > >
> > > Secondly, regarding your suggestion to further discuss the performance of downstream tasks, particularly in comparison to other methods, we believe this result is quite evident. Since our proposed SeaDiff outperforms other methods in both FID and YOLO Score, the generated images should theoretically offer the best data augmentation for downstream tasks. On the other hand, we have experimentally validated this. We used data generated by the contrastive method GeoDiffusion, which performs best in YOLO Score, for downstream task enhancement. The results, shown in the table below, indicate that the performance improvement for downstream detectors using data generated by GeoDiffusion is not as significant as with our method, further proving the effectiveness of SeaDiff in enhancing downstream tasks.
> > >
> > > | Data     | YOLO  | YOLO  | Libra-RCNN | Libra-RCNN | Boosting-RCNN | Boosting-RCNN | Faster-RCNN | Faster-RCNN |
> > > |------------|-------|-------|------------|------------|---------------|---------------|-------------|-------------|
> > > |            | mAP50 | mAP50-95 | mAP50     | mAP50-95   | mAP50         | mAP50-95      | mAP50       | mAP50-95    |
> > > | RUOD                 | 0.567      | 0.421         | 0.698            | 0.492               | 0.764                | 0.541                 | 0.692             | 0.468                |
> > > | RUOD + GeoDiffusion  | 0.555      | 0.408         | 0.691            | 0.481               | 0.715                | 0.508                 | 0.670             | 0.441                |
> > > | **RUOD + SeaDiff (Ours)**   | **0.668**  | **0.502**     | **0.771**        | **0.525**           | **0.781**            | **0.579**             | **0.718**         | **0.470**            |
> > >
> > > In summary, SeaDiff outperforms the comparison methods in both image quality and downstream task enhancement, fully demonstrating its great potential for practical applications.

---

### Official Review · Reviewer_5nFM · 2024-11-03

**Soundness:** 3
**Presentation:** 3
**Contribution:** 3
**Rating:** 5
**Confidence:** 5

**Summary:**

This paper presents SeaDiff, an effective approach for controlling appearance attributes in underwater image generation using diffusion models. The authors model three key attributes (luminance, dynamic range, and color cast) and propose a Symmetrical Parameter Control framework consisting of an Appearance Writer (A-Writer) and an Appearance Reader (A-Reader). Evaluated on the RUOD dataset, the method demonstrates significant improvements over existing approaches in both appearance control and image quality.

**Strengths:**

The paper addresses controllable image attribute editing, which is a fundamental challenge. The focus on precise control of specific attributes (luminance, dynamic range, and color cast) demonstrates a clear understanding of key image manipulation needs. I think the research direction would leverage a wide range of applications.

**Weaknesses:**

In method section, I think the paper's unique contributions appear limited, primarily focusing on transferring existing pre-trained priors to a specific field. The proposed model is essentially a fine-tuned version of the pre-trained Stable Diffusion model. While leveraging pre-trained models is common practice, the core contributions and motivations of this work should be more explicitly highlighted to distinguish it from existing approaches.

In Table 1 of the experimental section, there is no analysis of the model's parameters or inference latency, which is essential given the reliance on a computationally heavy pre-trained model. Such an analysis would provide valuable insights into the method's efficiency and practical applicability.

In Section 5.1, the paper acknowledges a bias in existing datasets, which consist of similar frames extracted from the same video. However, the simple re-division of the dataset does not adequately address this distribution bias. Additionally, all experiments were conducted exclusively on the RUOD dataset, raising concerns about the method's generalization capabilities to other underwater datasets.

Meanwhile, the experimental section highlights successful outcomes but lacks a systematic analysis of failure cases or the method's limitations. A thorough examination of these aspects is crucial for understanding the robustness and applicability of the proposed method in diverse scenarios.

**Questions:**

Underwater image attributes encompass more than just luminance, dynamic range, and color cast. Can the proposed method be adapted to handle additional attributes or support more extensive attribute editing? If so, how would this be achieved?

The results presented in the experimental section demonstrate the method's performance under relatively ideal underwater conditions. However, there is no testing under extreme lighting conditions or severe turbidity. How does the method handle extreme underwater conditions such as severe turbidity, low visibility, and uneven lighting?

How does the method perform on underwater datasets other than RUOD? Has any cross-dataset evaluation been conducted to assess its generalization capabilities?

---

> ### Author Response · Authors · 2024-11-21
>
> ### Weekness 1: unique contributions appear limited
>
> The core contribution of this paper lies not in the simple fine-tuning of the pre-trained Stable Diffusion model but in the innovative introduction of the A-Writer and A-Reader modules, which enable precise control over underwater scene attributes. The A-Writer module injects appearance attributes into the U-Net encoder via cross-attention mechanisms, significantly enhancing the visual consistency and realism of the generated images. The A-Reader module extracts feature maps from each layer of the U-Net decoder, predicts appearance attributes, and provides deep supervision to ensure the generated images align closely with the desired attributes.
>
> Ablation experiments (Table 2) further validate the critical role of these two modules in improving model performance. Specifically, integrating A-Writer reduces the A-MSE from 1359.2 to 397.8; adding A-Reader brings it down to 490.2; and when both are used together, the A-MSE further decreases to 375.1. This substantial error reduction underscores the pivotal role of A-Writer and A-Reader in enhancing the control and quality of underwater scene attribute generation.
>
> ### Weekness 2: no analysis of the model's parameters or inference latency
>
> We fully understand the importance of model parameter size and inference latency in evaluating the efficiency and practical value of the approach. We sincerely apologize for overlooking this aspect in our initial experiments. To address this, we have conducted additional tests and provided the results for model parameter size and inference latency. All inference latency measurements were conducted on a single NVIDIA RTX 3090, with the batch size set to 1 during inference.
>
> | **A-Writer** | **A-Reader** | **Param** | **Inf. Time** |
> | ------------ | ------------ | --------- | ------------- |
> |              |              | 1117M     | 3805ms        |
> | ✓            |              | 1211M     | 4455ms        |
> |              | ✓            | 1124M     | 3810ms        |
> | ✓            | ✓            | 1232M     | 4846ms        |
>
> The results show that the A-Writer module significantly enhances generative capability while adding only 94M parameters and 650ms of inference time, demonstrating high efficiency. The A-Reader module, with minimal cost (adding only 7M parameters and 5ms of inference time), effectively improves the precision of attribute control. When both modules are used together, the total overhead is 115M parameters and 1041ms of inference time, achieving a well-balanced trade-off between performance improvement and inference efficiency. This indicates that the design of the A-Writer and A-Reader modules strikes a reasonable balance between performance and efficiency, providing strong support for the practical application of the method.

---

> ### Author Response · Authors · 2024-11-21
>
> ### Weekness 3: re-division of the dataset & all experiments were conducted exclusively on the RUOD dataset
>
> In Section 5.1, we addressed the issue not of dataset distribution bias but of data leakage. Specifically, this leakage manifests in cases where frame A of a video appears in the training set while its subsequent frame B appears in the test set. This allows the model, when reasoning on frame B during testing, to rely on its prior exposure to frame A and produce predictions similar to A. Given the sequential relationship between A and B, this often leads to predictions for frame B that are overly close to its ground truth. To resolve this issue, we restructured the RUOD dataset by designating the first 10,000 images as the training set, ensuring that no similar images appear in the test set and effectively eliminating the risk of data leakage.
>
> Without this redivision, during model training, we observed that the model easily generates results that closely resemble those in the test set. This phenomenon persists even when only simple box annotations are provided. It highlights that the potential overlap or relationship (e.g., consecutive frames) between the training and test sets allows the model to leverage implicit information, resulting in predictions that align closely with the targets in the test set. This underscores the critical importance of resolving data leakage to maintain fairness and reliability in model evaluation.
>
> In terms of dataset selection, we investigated several underwater object detection datasets, such as UTDAC, DUO, UODD, and RUOD. Our evaluation revealed that RUOD is a composite dataset that integrates images from various sources, including UTDAC, DUO, and UODD. In contrast, datasets like UTDAC, DUO, and UODD are often collected from single locations (as research institutions frequently focus on specific marine areas), resulting in limited diversity within individual datasets. RUOD, as a composite dataset, offers significantly greater diversity, encompassing images from varied scenarios and conditions. For this reason, we chose RUOD for our experiments, leveraging its broad coverage and diversity.
>
> Additionally, to further validate the generalization capability of the proposed method, we conducted zero-shot quantitative experiments on the UTDAC dataset. Specifically, the model was trained without using any images or label information from UTDAC and was then tested directly on UTDAC’s test set to assess its performance in unseen scenarios. This experimental design effectively evaluates the model’s cross-dataset generalization ability, particularly its adaptability to data from different marine areas or with varying characteristics. The results demonstrate that our method significantly outperforms existing approaches on the UTDAC test set, further confirming its robustness and generalization capability in diverse underwater object detection tasks.
>
> | Method             | Input Res. | A-MSE ↓  | SSIM ↑ | PSNR ↑   | FID ↓     |
> |--------------------|------------|----------|--------|----------|-----------|
> | ReCo              | 256×256    | 1149.3   | 0.173  | 14.199   | 149.677   |
> | GLIGEN            | 512×512    | 2105.1   | 0.182  | 12.579   | 196.895   |
> | InstanceDiffusion | 512×512    | 2540.0   | 0.142  | 12.307   | 217.559   |
> | GeoDiffusion      | 256×256    | 658.8    | **0.617** | 16.903   | 144.576   |
> | LayoutDiffuse     | 512×512    | 2906.0   | 0.164  | 12.885   | 186.084   |
> | **SeaDiff (Ours)** | 256×256    | **83.7** | 0.604 | **19.752** | **88.230** |

---

> ### Author Response · Authors · 2024-11-21
>
> ### Weekness 4: analysis of failure cases
>
> In the experimental design of this study, we focused on demonstrating the method's control capabilities over various attributes, such as brightness, dynamic range, and color bias, to validate the advantages of SeaDiff in underwater image generation. It is true that the current version has limited discussion of failure cases. To address this, we have included typical failure case examples in the supplementary materials and analyzed the errors between generated images and target attributes to identify the potential limitations of SeaDiff.
>
> In our experimental analysis, we observed that while the proposed method performs exceptionally well in most scenarios, it also exhibits certain limitations and failure cases. These cases are crucial for understanding the method's applicability and identifying directions for improvement. Below are two typical failure cases:
>
> 1. **Conflict Between Control Capability and Image Quality**
>     When strict control over image attributes is applied, such as adjusting brightness, dynamic range, or color bias, the model sometimes prioritizes attribute consistency to such an extent that the overall image quality deteriorates. In such cases, the generated images may suffer from detail loss or texture blurring, particularly in tasks involving high dynamic range or complex color bias. This indicates that when there is a trade-off between controlling attributes and maintaining high image quality, the model tends to favor attribute consistency at the expense of image quality.
> 2. **Control Failures in Complex Scenes**
>     In more complex scenes, especially those containing a large number of bounding boxes (boxes), the model demonstrates limitations in accurately controlling the appearance attributes of each target. Specifically, in scenarios with a high number of targets, the generated images may exhibit inconsistencies in appearance attributes, such as some targets failing to meet the specified color or brightness requirements. This suggests that as scene complexity increases, the model's ability to independently control the attributes of each target diminishes, exposing certain limitations in handling complex scenes.
>
> These failure cases provide valuable insights for further improvement. For instance, optimizing the balance between attribute control and image quality, as well as enhancing fine-grained control capabilities in complex scenes, could further improve the robustness and applicability of the method.
>
> ### Questions 1: more extensive attribute editing
>
> Indeed, underwater image properties are not limited to brightness, dynamic range, and color bias. This paper focuses on these three key attributes for discussion. It is worth emphasizing that the proposed method not only effectively controls these three attributes but also has the potential to handle a wider range of image properties. This is because the A-Writer and A-Reader modules designed in this study essentially parameterize the control of the Diffusion image generation process, making them applicable to attributes beyond brightness, dynamic range, and color bias.
>
> The implementation of this extension is also straightforward. Both the input of the A-Writer and the output of the A-Reader are n-dimensional vectors, where n represents the number of attributes to be controlled. To include additional attributes, one only needs to adjust the dimensions of the A-Writer and A-Reader modules accordingly. This flexibility allows the proposed method to easily adapt to various scenarios, further enhancing its applicability and versatility.
>
> ### Question 2: handle extreme underwater conditions such as severe turbidity, low visibility, and uneven lighting
>
> We appreciate the reviewer’s attention to testing under extreme underwater conditions. Turbidity and visibility are indeed critical environmental attributes. However, in this study, we have not yet implemented specific control over the generation of these attributes. Our primary goal was to propose an attribute-controlled method, using brightness, dynamic range, and color bias as examples. These attributes are prevalent in underwater environments and are easily quantifiable, making them suitable as initial validation indicators for the model.
>
> It is important to emphasize that the SeaDiff framework is both generalizable and extensible, capable of accommodating the definition and control of other attributes. In future research, we plan to incorporate a wider variety of environmental attributes, such as turbidity, visibility, and lighting uniformity, and accordingly refine the attribute injection and supervision modules of A-Writer and A-Reader. We believe these extensions will significantly improve the method's applicability and robustness, enhancing its performance under extreme underwater conditions and enabling broader applications in more complex scenarios.

---

> ### Author Response · Authors · 2024-11-21
>
> ### Question 3: performance on underwater datasets other than RUOD
>
> To validate the generalization capability of the proposed method, we conducted zero-shot experiments on the UTDAC dataset, independent of RUOD. During training, no images or labels from UTDAC were used; the model was evaluated on the UTDAC test set only during testing to assess its adaptability to unseen scenarios. The results demonstrate that our method significantly outperforms existing approaches on the UTDAC test set, further confirming its robustness and generalization capability in underwater object detection tasks with high diversity across different marine environments.
>
> | Method             | Input Res. | A-MSE ↓  | SSIM ↑ | PSNR ↑   | FID ↓     |
> |--------------------|------------|----------|--------|----------|-----------|
> | ReCo              | 256×256    | 1149.3   | 0.173  | 14.199   | 149.677   |
> | GLIGEN            | 512×512    | 2105.1   | 0.182  | 12.579   | 196.895   |
> | InstanceDiffusion | 512×512    | 2540.0   | 0.142  | 12.307   | 217.559   |
> | GeoDiffusion      | 256×256    | 658.8    | **0.617** | 16.903   | 144.576   |
> | LayoutDiffuse     | 512×512    | 2906.0   | 0.164  | 12.885   | 186.084   |
> | **SeaDiff (Ours)** | 256×256    | **83.7** | 0.604 | **19.752** | **88.230** |

---

### Author Response · Authors · 2024-11-28

Dear Reviewers,

First, we would like to express our sincere gratitude for your thorough review and valuable feedback. We have given careful consideration to each of your suggestions, and in response, we have updated our paper and conducted additional experiments to further validate and improve our work. Below are the key updates we made to the experimental section:

1. **Analysis of Model Parameters and Inference Latency**: We have updated the analysis of model parameters and inference latency, exploring how different parameter settings impact the model's performance and efficiency. This update addresses the suggestions from Reviewer 5nFM and Cfjp.

2. **Zero-Shot Quantitative Experiments on the UTDAC Dataset**: In response to Reviewer 5nFM’s feedback, we have extended our validation of the zero-shot task and included new quantitative experiments on the UTDAC dataset to ensure the stability and effectiveness of the model in real-world applications.

3. **Validation of Generated Image Effectiveness and Enhancement of Downstream Object Detection Methods**: We have further validated the effectiveness of the generated images and, based on the feedback from Reviewers Cfjp, HkDY, and PBAu, enhanced the downstream object detection methods to improve the model's performance across various tasks.

The updates and additional experiments have been incorporated into the latest version of the paper, and the updated PDF file has been submitted for your review. We deeply appreciate the constructive feedback you provided, which has greatly helped us improve the quality and depth of our research. We hope these updates provide stronger support for our work. If you have any further suggestions or questions, we would be happy to continue the discussion and make any additional improvements.

Thank you once again for your time and review!

Best regards,
The Authors

---

### Meta-Review · Area_Chair_9yed · 2024-12-19

**Metareview:**

The paper addresses appearance discrepancies in underwater images by modeling three attributes: luminance, dynamic range, and color cast. It introduces the Appearance Writer (A-Writer) module that encodes these attributes and employs a cross-attention mechanism within a U-Net encoder to enhance realism and consistency in the images. Additionally, the Appearance Reader (A-Reader) module is introduced to analyze feature maps from the U-Net decoder, predicting appearance attributes and providing deep supervision to ensure alignment with expected appearance, ultimately improving image quality and fidelity.

***Strengths:***
- The paper targets the specific topic generation of controllable and realistic underwater images, that may have potential usage for underwater applications.
- The manuscript is well-structured and written clearly,
- The use of a pre-trained SD model for fine-tuning is a practical approach

***Weaknesses:***
- The novelty of the proposed method is limited which just  fine-tuning an existing model with minimal new contributions to the method itself.
- The paper lacks critical analysis regarding the model's efficiency. This part is added, but the efficiency is not an advantage for this method.

We thank the authors for their rebuttal and additional explanations and experiments provided during the review process. After careful consideration and discussion, we conclude the paper does not currently meet the threshold for acceptance in its present form. The manuscript does not sufficiently differentiate its approach from existing methods. The primary contribution appears to be an application of widely used techniques without significant modifications or novel insights. The research presented does not meet the high standards expected for significant impact or advancement.

**Additional Comments On Reviewer Discussion:**

The author has provided a rebuttal with additional explanations and experimental results. However, the reviewers still put a negative side towards this paper and they stated that the method is incremental and the results are not sufficiently strong. Given these concerens, we cannot accept this paper in this form.

---

### Decision · Program_Chairs · 2025-01-22

Reject